# The Activation of JAK/STAT3 Signaling and the Complement System Modulate Inflammation in the Primary Human Dermal Fibroblasts of PXE Patients

**DOI:** 10.3390/biomedicines11102673

**Published:** 2023-09-29

**Authors:** Christopher Lindenkamp, Ricarda Plümers, Michel R. Osterhage, Olivier M. Vanakker, Judith Van Wynsberghe, Cornelius Knabbe, Doris Hendig

**Affiliations:** 1Institut für Laboratoriums- und Transfusionsmedizin, Herz- und Diabeteszentrum Nordrhein-Westfalen, Universitätsklinik der Ruhr-Universität Bochum, 32545 Bad Oeynhausen, Germany; clindenkamp@hdz-nrw.de (C.L.); rpluemers@hdz-nrw.de (R.P.); mosterhage@hdz-nrw.de (M.R.O.); cknabbe@hdz-nrw.de (C.K.); 2Center for Medical Genetics, Ghent University Hospital, 9000 Ghent, Belgium; olivier.vanakker@ugent.be (O.M.V.); judith.vanwynsberghe@ugent.be (J.V.W.)

**Keywords:** pseudoxanthoma elasticum, JAK/STAT3, complement system, baricitinib

## Abstract

Previous studies revealed a link between inflammation and overactivation of the Janus kinase (JAK)/signal transducer and activator of transcription (STAT) signaling in syndromes associated with aging. Pseudoxanthoma elasticum (PXE), a rare autosomal-recessive disorder, arises from mutations in *ATP-binding cassette subfamily C member 6* (*ABCC6*). On a molecular level, PXE shares similarities with Hutchinson–Gilford progeria syndrome, such as increased activity of senescence-associated- beta-galactosidase or high expression of inflammatory factors. Thus, this study’s aim was the evaluation of activated STAT3 and the influence of JAK1/2-inhibitor baricitinib (BA) on inflammatory processes such as the complement system in PXE. Analysis of activation of STAT3 was performed by immunofluorescence and Western blot, while inflammatory processes and complement system factors were determined based on mRNA expression and protein level. Our results assume overactivation of JAK/STAT3 signaling, increased expression levels of several complement factors and high C3 protein concentration in the sera of PXE patients. Supplementation with BA reduces JAK/STAT3 activation and partly reduces inflammation as well as the gene expression of complement factors belonging to the C1 complex and C3 convertase in PXE fibroblasts. Our results indicate a link between JAK/STAT3 signaling and complement activation contributing to the proinflammatory phenotype in PXE fibroblasts.

## 1. Introduction

Pseudoxanthoma elasticum (PXE, OMIM 264800) is a genetic autosomal recessive multisystem disorder with a prevalence of 1:25,000 to 1:100,000 in the general population. The main characteristic of PXE is the ectopic calcification of connective tissue, particularly elastic fibers. Clinical manifestations of PXE are seen in the skin, eyes or cardiovascular system. Symptoms of patients with PXE in the early stage of the disease are yellow papules at flexural body sites, which coalesce, leading to sagging and wrinkled skin [1,2,3]. In addition, hyperpigmentation (*peau d’orange*), fracturing (angioid streaks) and bleeding in the calcified Bruch’s membrane finally leads to the loss of central vision [4,5]. Patients with PXE also show cardiovascular manifestations, including hypertension and peripheral arterial vascular disease [6,7].

The origin of PXE are mutations in the gene of *ATP-binding cassette subfamily C member 6* (*ABCC6*), resulting in a loss of function of the encoded ABC-transporter protein [8]. ABCC6 is expressed mainly in the liver and kidney but only in small amounts in peripheral tissues, such as skin [9]. Over 500 different mutations in the *ABCC6* gene have been identified in PXE patients to date, but the physiological substrate of ABCC6 remains unknown, which makes the development of an effective therapy difficult [10].

Our previous studies showed an association between PXE and premature aging [11]. Further, PXE shares some similarities on a molecular level with other age-associated diseases such as the Hutchinson–Gilford progeria syndrome (HGPS). A disturbed pyrophosphate (PPi) homeostasis with increased alkaline phosphatase activity, as well as a decreased ATP and PPi plasma level resulting in ectopic calcification, is characteristic of both diseases [12,13,14]. Dermal skin fibroblasts from patients with PXE or HGPS show high activity of senescence-associated beta-galactosidase (SA-β-gal) and increased protein expression of genes belonging to the senescence-associated secretory phenotype (SASP) [11,15,16,17]. Proteins of the SASP include monocyte chemoattractant protein-1 (MCP-1), chemokine (C-X-C motif) ligand 1 (CXCL1) and interleukin-6 (IL-6) [18].

As shown before, IL-6 is considered one of the most prominent proinflammatory cytokines and plays an important role in chronic inflammatory processes [19,20]. On the molecular level, IL-6 forms a complex with IL-6 receptor (IL-6R), which binds to glycoprotein 130 (gp130). Interaction between the IL-6-IL6R complex and gp130 results in the recruitment and activation of Janus kinase (JAK) and the following phosphorylation of the signal transducer and activator of transcription 3 (pSTAT3). Two monomers of active pSTAT3 form a dimer that translocates into the nucleus and activates gene expression [21,22]. JAK/STAT3 signaling induces expression of the SASP genes, such as *MCP-1* and *IL-6*, as well as the regulatory protein *suppressor of cytokine signaling* (*SOCS3*). *SOCS3* is highly expressed in inflamed tissue and reduces activation of JAK1/2 [23,24]. An activation of the JAK/STAT3 pathway can be seen in inflammatory diseases such as Behçet’s Disease as well as in age-related chronic inflammation and dysfunction [25,26]. As shown in the literature, blocking JAK/STAT3 signaling reduces inflammatory processes and cellular senescence in age-related processes and diseases [27,28].

It is known that, in addition to IL-6 signaling, STAT3 activation can also be induced by complement factor 3 (C3) [29]. C3 belongs to the complement system, which plays an important role in innate immunity and inflammation [30]. The complement cascade is activated during tissue injury by classical (C1q, 1s, 1r, 2 and 4), lectin (C2, 4 and mannose-binding lectin serine proteases) and/or alternative activation (C3, factor B and D). The formation of the C3-convertase and cleavage of C3 results in a proinflammatory environment and cellular lysis [31,32]. The complement cascade is regulated by different proteins, such as the C1-inhibitor encoded by the gene *SERPING1*, which inhibits the classical pathway by blocking the enzymatic activity of the C1-complex. The alternative pathway is regulated by complement factor H (CFH), which inhibits the cleavage of C3, a necessary step for complement activation [33,34]. Deregulation of the complement system and chronic inflammatory processes can be seen in age-related diseases, such as age-related macular degeneration (AMD) [35].

Because of the association between PXE, premature aging and inflammatory processes, we hypothesized that JAK/STAT3 signaling and the complement system might play a role in PXE pathogenesis. For a better understanding of inflammatory processes in PXE pathogenesis, we investigate the activation of STAT3 and the expression of the complement system in dermal fibroblasts of PXE patients (PXEF). Furthermore, we analyze the PXE phenotype after blocking JAK/STAT3 signaling by the specific inhibitor baricitinib (BA) to determine the role of JAK/STAT3 signaling and the complement cascade in inflammatory processes in PXE.

## 2. Materials and Methods

### 2.1. Experimental Design

This study was designed to evaluate the activation status of JAK/STAT3 signaling and the effect of blocking JAK/STAT3 signaling by specific JAK1/2 inhibitor BA on primary human dermal fibroblasts of PXE patients. Accordingly, PXEF and fibroblasts from sex- and age-matched healthy controls (NHDF) were cultivated with 10% lipoprotein-deficient fetal calf serum (LPDS). Additional treatment with BA was performed to evaluate the effects on PXEF. The activation status of JAK/STAT3 signaling and the effect of BA on this activation status were evaluated in PXEF by immunofluorescence, Western blot and mRNA expression of *STAT3* and *suppressor of cytokine signaling 3*. Furthermore, the effect of BA was determined by measuring the mRNA expression of factors that are involved in the generation of SASP, such as *IL-6*, *MCP-1* and *CXCL1*, as well as the protein concentration of IL-6 and MCP-1. Furthermore, the mRNA expression of complement system factors *complement factor 1r* (*C1r*), *1s* (*C1s*), *2* (*C2*), *C3*, *4A* (*C4A*), *FB* (*CFB*), *CFH* and *SERPING1* and protein concentration in the supernatant of C3, C3a and C1r under standard conditions and treatment with BA were determined. Additionally, C3 protein concentrations in sera from PXE patients and sex- and age-matched healthy controls were examined.

### 2.2. Patient Characteristics

The diagnosis of PXE in all patients was consistent with the consensus criteria, as reported previously [36]. Sera from sex- and age-matched healthy blood donors were used as a control. Sera from PXE patients (n = 43, 30 females and 13 males, age 44.4 ± 13.5 years) and controls (n = 41, 29 females and 12 males, age 44.4 ± 12.9 years) were examined for analysis.

### 2.3. Cell Culture and Treatment

Primary PXEF from patients were isolated from skin biopsies as described previously [37]. All characteristics are listed in Table 1. NHDF were purchased from the Coriell Institute for Medical Research (Camden, NJ, USA). All patients were informed and gave their consent to use the material for research purposes. This study was carried out in accordance with the Declaration of Medicine, Ruhr University of Bochum (registry no. 32/2008, approval date 3 November 2008).

The PXEF and NHDF were cultivated in Dulbecco’s modified essential medium (Gibco, Thermo Fisher Scientific, San Diego, CA, USA) containing 10% fetal calf serum (FCS, Biowest, Aidenbach, Germany), 2% L-glutamine (200 mM) (PAN, Biotech, Eidenach, Germany) and 1% antibiotic/antimycotic solution (PAA Laboratories, Pasching, Austria). After reaching 85–95% confluence, fibroblasts were sub-cultivated. Fibroblasts between passages 8 and 12 were used in this study. Biological samples were prepared in triplicate. In an experimental context, fibroblasts were seeded in a final density of 177 cells/mm^2^ in 60 mm culture dishes (BD Falcon, Schaffhausen, Switzerland) or on coverslips (Ø 18 mm) coated with 5 µg/cm^2^ rat collagen (Ibidi, Graefelfing, Germany) for immunofluorescence experiments and were cultivated in medium supplemented with 10% FCS. After 24 h, the medium was changed to Dulbecco’s modified essential medium supplemented with 10% LPDS and 0.01% (*v*/*v*) dimethyl sulfoxide (DMSO; vehicle), with BA in a final concentration of 1 µM or for immunofluorescence with IL-6 (50 ng/mL) or BA (1 µM) and IL-6 (50 ng/mL). Fibroblasts were cultivated for an additional 72 h, prepared for further experiments and analyzed.

### 2.4. Delipidation of FCS

The preparation of LPDS was carried out according to [39]. In brief, 50 mL of FCS was treated with 1 g Cab-o-sil (Sigma, Taufkirchen, Germany) at 4 °C overnight. After incubation, the mixture was centrifuged at 10,000× *g* for 1 h at 4 °C. The supernatant was transferred into a new tube and used directly or stored at −20 °C until further use. The LPDS was sterile filtered (0.2 µm, Sartorius, Göttingen, Germany) before preparation of medium. After delipidation, the concentration of free cholesterol was lowered by about 78%, low-density lipoprotein by about 95% and high-density lipoprotein by about 57%, but the triglyceride concentrations remained unchanged, as described previously [40].

### 2.5. Isolation of Nucleic Acid

A commercially available NucleoSpin RNA Kit (Macherey-Nagel, Düren, Germany) was used for the isolation of RNA. DNA isolation was carried out by using a NucleoSpin Blood Extraction Kit (Macherey-Nagel, Düren, Germany). Isolation procedure was performed as described by the manufacturer. For both, the procedures were determined on a NanoDrop2000 spectrophotometer (Peqlab, Erlangen, Germany).

### 2.6. Gene Expression Analysis

A total of 1 µg RNA was transcribed into cDNA by using the SuperScript II Reverse Transcriptase (Thermo Fisher Scientific, San Diego, CA, USA), as described by the manufacturer. Regarding all gene expression measurements, 2.5 µL cDNA (1:10 diluted in water), 0.25 µL forward and reverse primer (Biomers, Ulm, Germany) with a final concentration of 25 µM, 2 µL water and 5 µL LightCycler 480 SYBR Green I Master reaction mixture (Roche, Penzberg, Germany) were used. The protocol for quantitative real-time PCR (qPCR) is shown in Table 2. The mRNA expression of the genes *ß-actin* (*ACTB*), *glyceraldehyde-3-phosphatase-dehydrogenase* (*GAPDH*) and *ß2-microglobulin* (*ß2M*) were used for normalization. Finally, a melting curve was measured after amplification for the analysis of purity. Three technical replicates were performed for each biological sample. All measurements were carried out by using a LightCycler480 (Roche, Penzberg, Germany). The ΔΔCt method, considering PCR efficiency, was used for the calculation of relative mRNA expression. All sequences of the primer used for qPCR are listed in Appendix A.

### 2.7. Immunofluorescence and Western Blot Analysis of Active pSTAT3

Fibroblasts were analyzed via immunofluorescence for the evaluation of active pSTAT3. After treatment with BA, cells were washed once with 1× PBS and fixed with 4% formaldehyde (4% Roti Histofix, Roth, Karlsruhe, Germany) for 10 min. Afterwards, fibroblasts were washed three times for 5 min each and permeabilized in cold methanol (Merck, Darmstadt, Germany). After three washing steps, each for 5 min, unspecific binding sites were blocked for 1 h at room temperature in 1× PBS supplemented with 5% goat serum (Sigma-Aldrich, St. Louis, MO, USA) and 0.3% Triton X-100 (Sigma-Aldrich, St. Louis, MO, USA). After the blocking step, fibroblasts were incubated with primary anti-pSTAT3 (pTyr705) antibody (1:200, 9145S, Cell Signaling Technology, Danvers, MA, USA) overnight at 4 °C. The next day, the fibroblasts were washed three times for 5 min each with 1× PBS and incubated with secondary goat anti-rabbit antibody (1:200, ab150078, Abcam, Cambridge, United Kingdom) for 1 h at room temperature. After an additional three washing steps (each 5 min), fibroblasts were counterstained with 4′,6-diamidino-2-phenylindole (DAPI). Finally, the coverslips were washed three times in 1× PBS and mounted with ProLong Diamond Antifade Mountant (Thermo Fisher Scientific, San Diego, CA, USA). Six pictures per cell culture condition were analyzed for the assessment of immunofluorescence by using the software ImageJ 1.53. Briefly, the area of the nuclei was bordered, and the fluorescence intensity was measured and relativized to the area.

In addition to immunofluorescence analysis, active pSTAT3 was analyzed by Western blot. Fibroblasts were cultivated on 60 mm culture dishes as described above. After cultivation, fibroblasts were washed once in 1× PBS and lysed in radioimmunoprecipitation assay buffer (Thermo Fisher Scientific, San Diego, CA, USA) supplemented with 1× protease inhibitor cocktail (Sigma-Aldrich, Taufkirchen, Germany) and 1× phosphatase inhibitor cocktail (Cell Signaling Technology, Danvers, MA, USA). Lysates were centrifuged at 2000× *g* for 5 min and stored at −80 °C for further experiments. An amount of 20 µg of protein and 5 µL multicolor broad range protein ladder (Thermo Fisher Scientific, San Diego, CA, USA) were separated by using 8–16% sodium dodecyl sulfate-polyacrylamide gel electrophoresis (Thermo Fisher Scientific, San Diego, CA, USA). The protein blots were transferred on polyvinylidene fluoride membrane (Thermo Fisher Scientific, San Diego, CA, USA) in a Mini Gel Tank (Invitrogen, Waltham, MA, USA). After transfer, membranes were blocked in 5% skimmed milk in Tris-buffered saline supplemented with 0.1% (*v*/*v*) Tween 20 for 1 h. Thereafter, membranes were incubated with primary antibodies against pSTAT3 in 5% skimmed milk (1:2000, Cell Signaling Technology, Danvers, MA, USA) or GAPDH (1:2000, Abcam, Cambridge, UK) overnight at 4 °C. The next day, membranes were washed three times in Tris-buffered saline with Tween 20 and incubated for 1 h at room temperature with secondary antibodies conjugated with horseradish peroxidase against rabbit (1:5000, Abcam, Cambridge, UK) or mouse (1:2000, Cell Signaling Technology, Danvers, MA, USA). After three additional washing steps, the protein band detection was performed using an enhanced chemiluminescence detection system (Thermo Fischer Scientific, San Diego, CA, USA). Band densitometry-based quantification of three biological replicates was carried out by using ImageJ 1.53.

### 2.8. Immunoassays for the Evaluation of IL-6, MCP-1, C3, C3a and C1r Concentration in Cell Culture Supernatant and C3 Concentration in Sera from PXE Patients

Analysis of the protein concentration of IL-6 in cell culture supernatants was carried out by using the Immunoanalyzer Cobas e411 (Roche, Basel, Switzerland). Determination of MCP-1, C3, C3a and C1r in cell culture supernatants and in sera of PXE patients were conducted by using a commercially available enzyme-linked immunosorbent assay (MCP-1: R&D Systems, Abingdon United Kingdom; C3 and C1r: Abcam, Cambridge United Kingdom; C3a: RayBiotech, Peachtree Corners, GA, USA). The protein concentrations measured were normalized to DNA content.

### 2.9. Statistical Analysis

All data are shown as mean ± standard error for the qPCR, protein concentrations in cell culture supernatants and quantification of immunofluorescence and Western blot. GraphPad Prism 9.0 was used, and the non-parametric two-tailed Mann–Whitney U test was performed for statistical analysis. *p*-values of 0.05 or less were considered statistically significant.

## 3. Results

### 3.1. Blocking JAK/STAT3 Signaling Reduced the Protein Level of Active pSTAT3 in PXE Fibroblasts

The level of active pSTAT3 was evaluated by immunofluorescence and Western blot to determine the basal activation of STAT3 in PXEF and NHDF and the effect of BA on pSTAT3 expression. Accordingly, fibroblasts were treated with DMSO as a vehicle (untreated), with BA for blocking JAK/STAT3 signaling, with IL-6 to induce JAK/STAT3 signaling (positive control), or in combination. Analysis of the signal (intensity/area) for active pSTAT3 showed a significant 28-fold higher signal of active pSTAT3 in untreated PXEF (intensity/area: 4.5 ± 1.81) compared to untreated NHDF (intensity/area: 0.2 ± 0.05). Treatment with BA reduced the level of active pSTAT3 in PXEF by 98% (intensity/area 0.1 ± 0.02) and for NHDF by 50% (intensity/area: 0.1 ± 0.02) compared to untreated fibroblasts. There was no significant difference in the level of active pSTAT3 in IL-6-treated PXEF (absolute intensity/area: 6.3 ± 1.16) compared to untreated PXEF, but a significant 84-fold higher signal could be seen for IL-6-treated NHDF (intensity/area: 13.4 ± 1.96) in contrast to untreated NHDF. Furthermore, a significantly reduced signal for active pSTAT3 was shown in BA + IL-6-treated PXEF (intensity/area: 0.04 ± 0.01, reduction by 98%) and NHDF (intensity/area: 0.05 ± 0.01, 99%) in comparison to the corresponding untreated fibroblasts (Figure 1A,B).

Analysis via Western blot also showed similar effects. A significant 14-fold higher signal for pSTAT3 (normalized to GAPDH protein expression) could be detected in untreated PXEF (1.5 ± 0.26 AU) compared to untreated NHDF (0.1 ± 0.03 AU). Additionally, a significant reduction of the pSTAT3 signal in BA-treated (0.08 ± 0.02 AU) compared to untreated PXEF by 94% was detected. No difference in the pSTAT3 signal could be seen in untreated NHDF compared to BA-treated NHDF (0.09 ± 0.01 AU) and to BA-treated PXEF (Figure 1C,D, Appendix A).

The gene expressions of *STAT3* and *SOCS3* were analyzed in addition to immunofluorescence and Western blot. As described in Figure 2, there was a significant increase of mRNA expression of *STAT3* and *SOCS3* in untreated PXEF (*STAT3:* 1.6 ± 0.15, 1.5-fold; *SOCS3*: 5.1 ± 0.84, 5-fold) compared to untreated NHDF. Gene expression of both targets was significantly decreased in PXEF (*STAT3:* 0.9 ± 0.06, 42%; *SOCS3:* 0.4 ± 0.13, 91%) after treatment with BA, while only *SOCS3* expression was significantly decreased by 42% in NHDF after BA treatment compared to corresponding untreated fibroblasts (Figure 2A,B).

### 3.2. Blocking JAK/STAT3 Pathway Partly Reduces Senescence-Associated Secretory Phenotype in PXE Fibroblasts

To further evaluate the role of JAK/STAT3 signaling on inflammatory processes in PXE pathogenesis, members of the SASP were analyzed in PXEF.

The measurement of IL-6 gene expression (PXE: 11.6 ± 2.02, 11-fold) and the protein level in supernatants (PXE: 227.0 ± 86.7 pg/µL/(µg DNA), 19-fold; NHDF: 12.1 ± 3.04 pg/µL/(µg DNA)) showed a significant increase in untreated PXEF compared to untreated NHDF. A treatment with BA revealed no significant differences in *IL-6* gene expression (8.6 ± 1.5 pg/µL/(µg DNA)) and protein level (190.0 ± 71.7 pg/µL/(µg DNA)) in PXEF in comparison to untreated PXEF. Only a significant reduction of *IL-6* gene expression by 16% was for BA-treated NHDF, but there was no significant difference in the IL-6 protein level (untreated: 12.0 ± 3.0, BA: 9.0 ± 2.6 pg/µL/(µg DNA)) compared to untreated NHDF (Figure 3A,B).

A significantly higher *MCP-1* expression could be detected in untreated PXEF compared to untreated NHDF (PXE: 7.9 ± 1.1, 8-fold). MCP-1 protein concentration was also increased in PXEF supernatants (PXE: 1111.0 ± 275.7 pg/µL/(µg DNA, 6-fold; con: 178.0 ± 40.4 pg/µL/(µg DNA))). The BA-treated PXEF showed a significant reduction of *MCP-1* gene expression by 68% (2.6 ± 0.36) and a tendential reduction of the MCP-1 protein level by 64% (401.4 ± 98.5 pg/µL/(µg DNA)) in comparison untreated PXEF. For BA-treated NHDF, there was a significant reduction in the *MCP-1* gene expression by 52% (0.49 ± 0.03) and on protein level by 51% (87.9 ± 19.5 pg/µL/(µg DNA)) compared to untreated NHDF (Figure 3C,D).

In addition to the gene expression of *IL-6* and *MCP-1*, the mRNA expression of *CXCL1* was measured. *CXCL1* gene expression was significantly increased in PXEF (5.3 ± 1.5; 5-fold) in contrast to NHDF (0.98 ± 0.05) (Figure 3E). A treatment with BA significantly reduced the *CXCL1* expression in NHDF by 13% (0.85 ± 0.12) in contrast to untreated NHDF. The *CXCL1* mRNA expression was only tendentially reduced by 33% (3.5 ± 0.82) in BA-treated PXEF in comparison to untreated PXEF (Figure 3E).

### 3.3. Increased Gene Expression and Protein Level of Complement Factors in PXE Fibroblasts Are Partially Affected by Blocking JAK/STAT3 Signaling

To determine the role of the complement system and its association with JAK/STAT3 signaling in PXE pathogenesis, factors of the complement system were analyzed in PXEF.

The gene expression (PXE: 11.4 ± 1.3, 11-fold) and protein concentration in supernatants (PXE: 1.1 ± 0.30 (ng/mL)/µg DNA, 4-fold; con: 0.27 ± 0.07 (ng/mL)/µg DNA) of C1r were significantly increased in untreated PXEF in comparison to untreated NHDF (Figure 4A,B). BA-treated PXEF display a significant reduction in gene expression by 85% (11.4 ± 1.3) and in protein concentration by 57% (1.1 ± 0.31 (ng/mL)/µg DNA) compared to untreated PXEF. BA treatment in the NHDF reduced the *C1r* mRNA expression significantly by 58% in contrast to untreated NHDF (0.5 ± 0.15) (Figure 4A,B).

The gene expression of *C1s* (PXE: 3.9 ± 0.46, 3.7-fold), *C2* (PXE: 12.5 ± 3.64, 11.9-fold) and *CFB* (PXE: 3.2 ± 0.49, 3-fold) were significantly increased; additionally, *C4A* (PXE: 1.7 ± 0.31, 1.6-fold) expression was tendentially increased in untreated PXEF compared to untreated NHDF (Figure 4C–F). The addition of BA to PXEF reduced *C1s* (1.7 ± 0.25) and *C4A* (0.81 ± 0.05) gene expression significantly by 57% and 51% in contrast to untreated PXEF. The mRNA expression of *C2* (3.9 ± 0.72) and *CFB* (1.8 ± 0.17) were only tendentially reduced by 68% and 45% in BA-treated PXEF compared to untreated PXEF. After BA supplementation, the gene expression of *C1s* (0.63 ± 0.06) and *C2* (0.84 ± 0.1) was significantly decreased by 40% and 19.7%, respectively, in NHDF in comparison to the untreated NHDF (Figure 4C–F).

In addition, the gene expression of complement inhibitors *CFH* and *SERPING1* was measured, and the effect of BA treatment was analyzed (Figure 4G,H). The gene expression of *CFH* was significantly reduced by 66% (PXE: 0.4 ± 0.02), while *SERPING1* gene expression was significantly enhanced (PXE: 3.3 ± 0.53, 3-fold) in untreated PXEF compared to untreated NHDF (Figure 4G,H). A supplementation of PXEF with BA significantly increased the *CFH* gene expression (0.60 ± 0.07, 1.7-fold) and reduced *SERPING1* gene expression tendentially by 30% (2.3 ± 0.21) in comparison to untreated PXEF. No significant difference for both gene expressions could be detected for BA-treated in contrast to untreated NHDF.

Next, we analyzed downstream factors of the complement system, such as C3 and C3a, the proteolytic product of C3 convertase, to determine the complement system activation level and its association with JAK/STAT3 signaling in PXEF.

The gene expression (PXE: 66.4 ± 16.5, 82-fold) and protein concentration (PXE: 5.3 ± 2.3 (ng/mL)/µg DNA, 35-fold; con: 0.15 ± 0.05 (ng/mL)/µg DNA) in the supernatant were significantly increased for C3 in untreated PXEF compared to untreated NHDF (Figure 5A,B). The BA treatment of PXEF and NHDF had no effect on the gene expression and protein concentration in supernatants of C3 in contrast to the untreated fibroblasts (Figure 5A,B). A significant increase of proteolytic product C3a could also be seen in untreated PXEF (0.21 ± 0.06 (ng/mL)/µg DNA) in comparison to untreated NHDF (0.02 ± 0.01 (ng/mL)/µg DNA). No differences in C3a protein concentration in supernatant between BA-treated PXEF and NHDF in contrast to corresponding untreated fibroblasts were detected.

To further analyze the systemic role of the complement system, C3 protein concentrations were measured in sera from PXE patients and healthy controls. We detected a significantly increased protein concentration of C3 in sera of PXE patients compared to healthy controls (PXE: 3.5 ± 0.19 mg/mL, 1.4-fold; con: 2.55 ± 0.18 mg/mL; Figure 5D).

## 4. Discussion

Our previous studies revealed a link between premature cellular senescence and the increased production of inflammatory factors of the SASP in PXEF [11,15]. One of these SASP factors is IL-6, which is highly expressed by PXEF, as shown before [15]. As described before, IL-6 forms a complex with IL-6R and activates the JAK/STAT3 pathway, which is linked to proinflammatory processes in age-related and chronic inflammatory diseases [21,25,26]. Thus, we hypothesized that JAK/STAT3 signaling might play a role in PXE pathogenesis.

For this, we evaluated the activation status of JAK/STAT3 signaling by analyzing active pSTAT3 on protein level and *STAT3* mRNA level. We found a significant increase of active pSTAT3 in nuclei and *STAT3* gene expression in PXEF. Only active pSTAT3 is able to translocate into the nucleus [22], indicating a permanent activation of JAK/STAT3 signaling in PXEF. Treatment with BA reduced the level of active pSTAT3 and *STAT3* expression in PXEF but not in NHDF. As shown in different studies, BA is a specific JAK inhibitor, preventing phosphorylation and activation of STAT3 [27,41], which explains the reduced level of active pSTAT3 and reduced gene expression of *STAT3* in BA-treated PXEF. Interestingly, there was no difference in the level of active pSTAT3 in untreated and IL-6-treated PXEF. This assumes an activation of JAK/STAT3 signaling in untreated PXEF, which could be reached by high IL-6 levels.

Furthermore, we showed an increased gene expression of *SOCS3,* a negative regulator of JAK/STAT3 signaling in PXEF [23], which inhibits the activity of JAK and, consequently, the activation and phosphorylation of STAT3 [42]. Previous studies revealed an association between high *SOCS3* expression, STAT3 activation and inflammation [24]. Thus, increased *SOCS3* gene expression in PXEF could be a sign of local inflammatory processes caused by activation of JAK/STAT3 signaling. Blocking JAK/STAT3 signaling correlated with a decrease of *SOCS3* gene expression in PXEF, supporting this hypothesis.

To evaluate the role of JAK/STAT3 signaling in chronic inflammatory processes in PXE, we next determine the effect of blocking JAK/STAT3 signaling on gene expression and protein level of SASP factors, which were shown to be dysregulated in PXE [11].

Examination of IL-6 revealed an increased gene expression and protein level in the supernatant of untreated PXEF in comparison to NHDF, which confirms our previous study [15]. Blocking JAK/STAT3 signaling did not affect the *IL-6* gene expression and the IL-6 protein level in supernatant in PXEF. IL-6 induces its own expression by JAK/STAT3 signaling [43], so blocking this pathway should decrease the IL-6 gene expression and protein secretion, as indicated by further studies [27,44]. Our results indicate high activation of JAK/STAT3 signaling in PXEF as a result but probably not as the origin of high IL-6 production, as neither BA concentration nor duration of treatment was sufficient to counteract the IL-6 overexpression.

In the case of MCP-1 and CXCL1, we showed increased gene expression and protein levels in untreated PXEF. As both chemokines play a pivotal role in chronic inflammatory processes [45,46], this supports the hypotheses of inflammatory processes in PXE. Further, it is known that CXCL-1 and MCP-1 recruit monocytes and other inflammatory cells to the source of inflammation [46,47]. Therefore, their increased expression by PXEF might reflect an enhanced infiltration of monocytes into tissue [48]. In fact, there are several studies describing the infiltration of immune cells into tissue in PXE patients pointing towards chronic inflammation in PXE patients supporting this hypothesis. Blocking JAK/STAT3 signaling via BA reduces MCP-*1* expression but not *CXCL1* gene expression in PXEF. Several studies have already shown that IL-6 stimulates *MCP-1* expression via JAK/STAT3 signaling [49,50,51,52], so blocking JAK/STAT3 signaling should decrease MCP-1 expression, which agrees with our results. However, BA treatment seemed to be limited and did not reduce the expression of MCP-1 to the level of healthy NHDF, which might indicate the need for higher BA concentration or prolonging the duration of treatment.

Besides investigating the effect of blocking JAK/STAT3 signaling on SASP factors, we focused on the complement system and its role in PXE pathogenesis. Therefore, we analyze the influence of JAK/STAT3 inhibition on the expression of different complement factors.

First, we analyzed complement factors initiating the complement cascade. We found an increased gene expression of *C1r C1s* and *CFB*, as well as an increased C1r protein concentration in supernatants in untreated PXEF compared to NHDF. C1s and C1r are components of the C1 complex, part of the classical complement cascade, while CFB plays an important role in the activation of the alternative complement cascade [31], but both pathways result in hydrolyzation of C3. The high expression of factors belonging to the C1 complex and *CFB* might indicate the activation of a classic complement cascade as well as an alternative complement cascade, which is initiated by the formation of the C1 complex or spontaneous hydrolyzation of C3 [53]. A previous study showed a correlation between the activity of the C1 complex and increased tissue inflammation [54]. Riihilä et al. also showed high gene expression of *C1r* and *C1s* in keratinocyte-derived cutaneous squamous carcinoma cells, which is associated with chronic inflammation, supporting the hypothesis of possible chronic inflammatory processes in PXEF [55]. Blocking JAK/STAT3 via BA reduced the expression of C1r and *C1s* in PXEF, indicating a regulation of classic complement cascade but not alternative cascade via JAK/STAT3 signaling in PXEF.

Next, we analyzed the gene expression of *SERPING1*, an inhibitor of the classic complement cascade, by blocking the C1 complex, and *CFH*, an inhibitor of the alternative complement cascade. Regarding *SERPING*, we discovered a significantly high mRNA expression in PXEF, which is not affected by BA supplementation. As shown in the literature, *SERPING1* belongs to the acute phase proteins, which are highly expressed in inflamed tissues and correlate positively with the gene expression of C1 factors [33]. Therefore, an increased mRNA expression indicates a high occurrence of proteins belonging to the C1 complex, supporting the hypothesis of an active complement cascade via the classical pathway. Regarding *CFH* gene expression, we identified a reduction in PXEF, while the *CFH* expression was upregulated after supplementation with BA. CFH is one of the key regulators of the complement system. Previous studies revealed a connection between a potential risk of AMD and polymorphisms in *CFH,* resulting in a downregulation of CFH expression and overactivation of the complement system [34,56]. Consequently, there is a similarity between PXE and AMD; thus, our previous studies analyzed CFH as a potential secondary risk factor for PXE. However, no genotype–phenotype correlation was identified in PXE patients [57]. Thus, a decreased expression of *CFH* and increased gene expression and protein level secretion of complement factors could indicate the missing regulation of the complement system, resulting in complement activation in PXE, which could partially be regulated by inhibiting JAK/STAT3 signaling.

We then analyzed the expression of proinflammatory C3, which is a central protein in the complement cascade. We discovered a strongly increased expression of C3 and C3a in untreated PXEF. The increased level of C3a, a product of C3 convertase, additionally indicates the high activity of the complement system in PXEF. Blocking JAK/STAT3 signaling did not affect the C3 gene expression or protein level of C3 and indicated no regulation of *C3* expression by STAT3 in PXEF. Yuan et al. described that JAK/STAT3 signaling is induced by *C3* overexpression [29], so increased *C3* expression could induce JAK/STAT3 signaling in addition to IL-6-mediated signaling in PXEF. We also analyzed the C3 concentration in sera from PXE patients to determine a possible link between local C3 production and systemic C3 circulation. We found significantly increased C3 protein levels in sera from PXE patients. High C3 levels could also be seen in patients with inflammatory bowel disease, which is associated with chronic inflammation [33]. Therefore, increased C3 levels in sera could assume chronic inflammation and an activated complement cascade not only locally in PXEF but also systemically in PXE patients.

To investigate the role of JAK/STAT3 signaling on downstream processes of the complement cascade in PXEF, we analyzed gene expression of *C2* and *C4A,* which form the C3 convertase together with C3b after cleavage of C3 by C1 complex [31]. We found an increased gene expression of both targets in PXEF that was reduced after blocking JAK/STAT3 signaling in PXEF. These results indicate a regulatory role of JAK/STAT3 not only in the initiation of the complement system but also in the downstream processes of the complement cascade.

## 5. Conclusions

In conclusion, this is the first study linking *ABCC6* deficiency in PXEF to a permanent activation of JAK/STAT3 signaling and activation of the complement cascade. Our results indicate that JAK/STAT3 activation partly modulates chronic inflammatory processes, but this seems to be limited in PXEF. Thus, further studies are necessary to evaluate the association of IL-6 to the permanent activation of JAK/STAT3 signaling. Other JAK/STAT signaling pathways and downstream factors of the complement system should be analyzed in PXEF to uncover the origin of chronic inflammation observed in PXE.

## Figures and Tables

**Figure 1 biomedicines-11-02673-f001:**
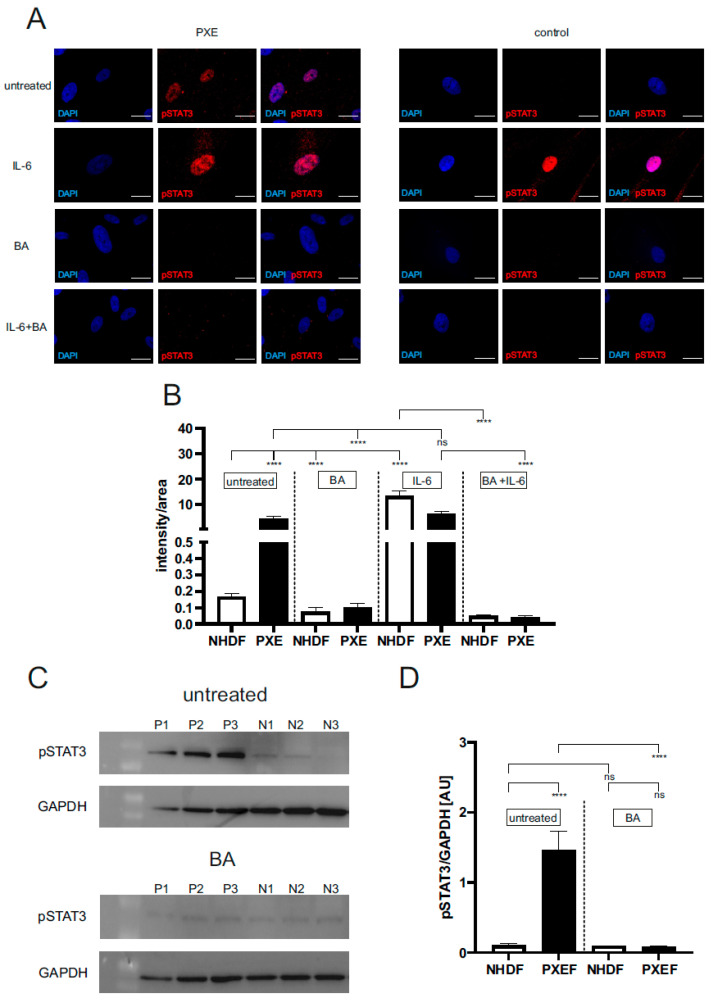
Immunofluorescence and Western blot of pSTAT3 (pTyr705). Dermal fibroblasts from patients with pseudoxanthoma elasticum (PXEF) (n = 4) and normal human dermal fibroblasts (NHDF) (n = 4) were cultivated for 72 h in medium with 10% lipoprotein deficient fetal calf serum (LPDS). Fibroblasts were treated only with DMSO (untreated), 1 µM baricitinib (BA), 50 ng/mL interleukin-6 (IL-6) or together (BA + IL-6). (**A**) Immunofluorescence analysis of pSTAT3 (red); cells were counterstained with DAPI (blue). (**B**) Quantification of pSTAT3 (intensity/area) in PXEF (black) and NHDF (white). (**C**) Western blot analysis of pSTAT3 expression of PXE (P) and NHDF (N). (**D**) Quantification of pSTAT3 normalized on GAPDH expression shown in arbitrary unit (AU) in PXE (black) and control (white) fibroblasts. Representative images are shown at 100× magnification (A, scale bar 15 µm). Data are shown as mean ± SEM. **** *p* ≤ 0.0001, ns *p* > 0.05.

**Figure 2 biomedicines-11-02673-f002:**
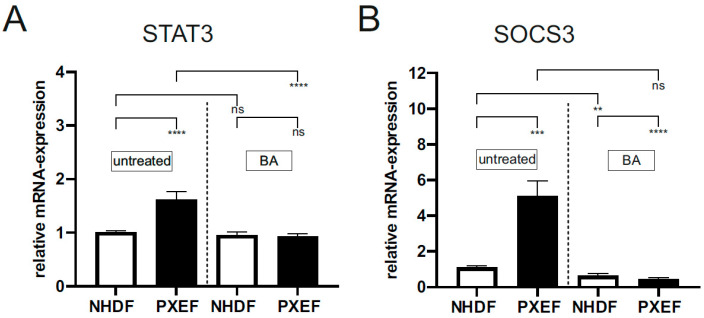
Relative *STAT3* and *SOCS3* mRNA expression in PXE and control fibroblasts. Fibroblasts from PXE patients (PXEF) and healthy donors (NHDF) were cultivated for 72 h in 10% LPDS medium with DMSO as a vehicle (untreated) or 1 µM BA. (**A**) Relative *STAT3* mRNA expression of PXEF (black) (n = 4) and NHDF (white) (n = 4). (**B**) Relative *SOCS3* mRNA expression of PXEF (black) (n = 4) and NHDF (white) (n = 4). Data are shown as mean ± SEM. **** *p* ≤ 0.001, *** *p* ≤ 0.002, ** *p* ≤ 0.01, ns *p* > 0.05.

**Figure 3 biomedicines-11-02673-f003:**
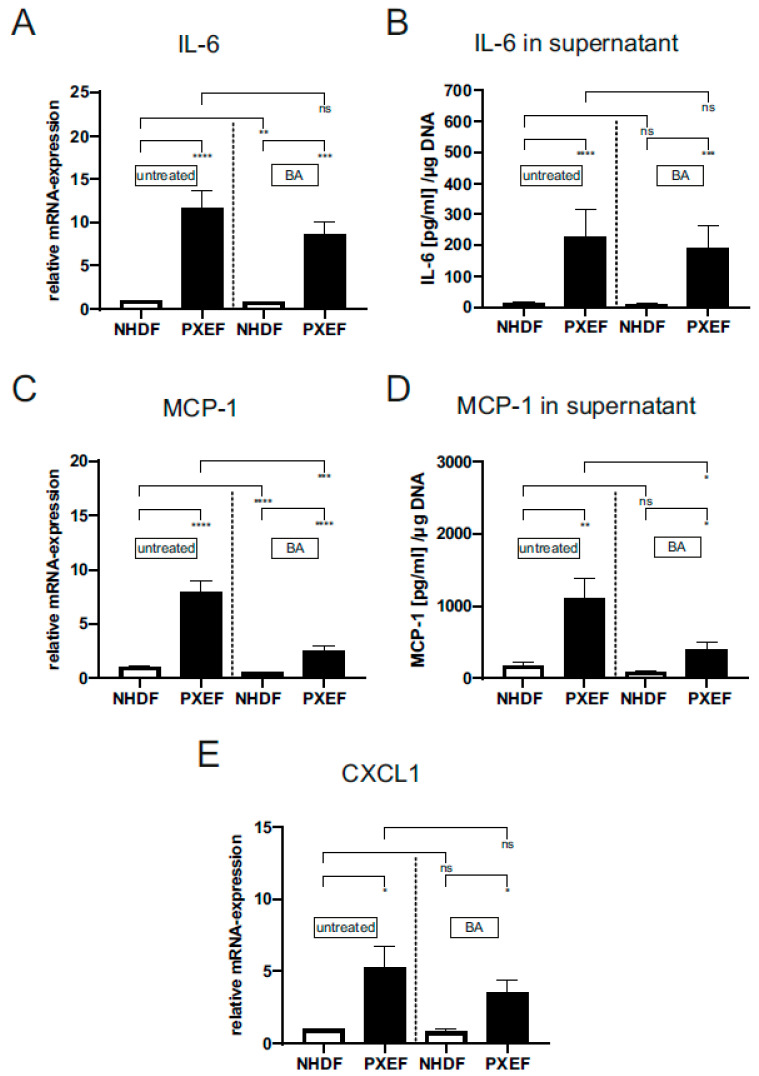
Relative mRNA expression of *IL-6*, *MCP-1*, *IGFBP3*, *CXCL1* and protein level of IL-6 and MCP-1 in PXE and control fibroblasts. Fibroblasts from PXE patients (PXEF) and healthy control donors (NHDF) were cultivated for 72 h in 10% LPDS medium with DMSO as vehicle (untreated) or 1 µM BA. Gene expression and protein level of (**A**,**C**) IL-6, (**B**,**D**) MCP-1 and (**E**) *CXCL1* in PXEF (black) (n = 4) and NHDF (white) (n = 4). Data are shown as mean ± SEM. **** *p* ≤ 0.001, *** *p* ≤ 0.002, ** *p* ≤ 0.01, * *p* ≤ 0.05, ns *p* > 0.05.

**Figure 4 biomedicines-11-02673-f004:**
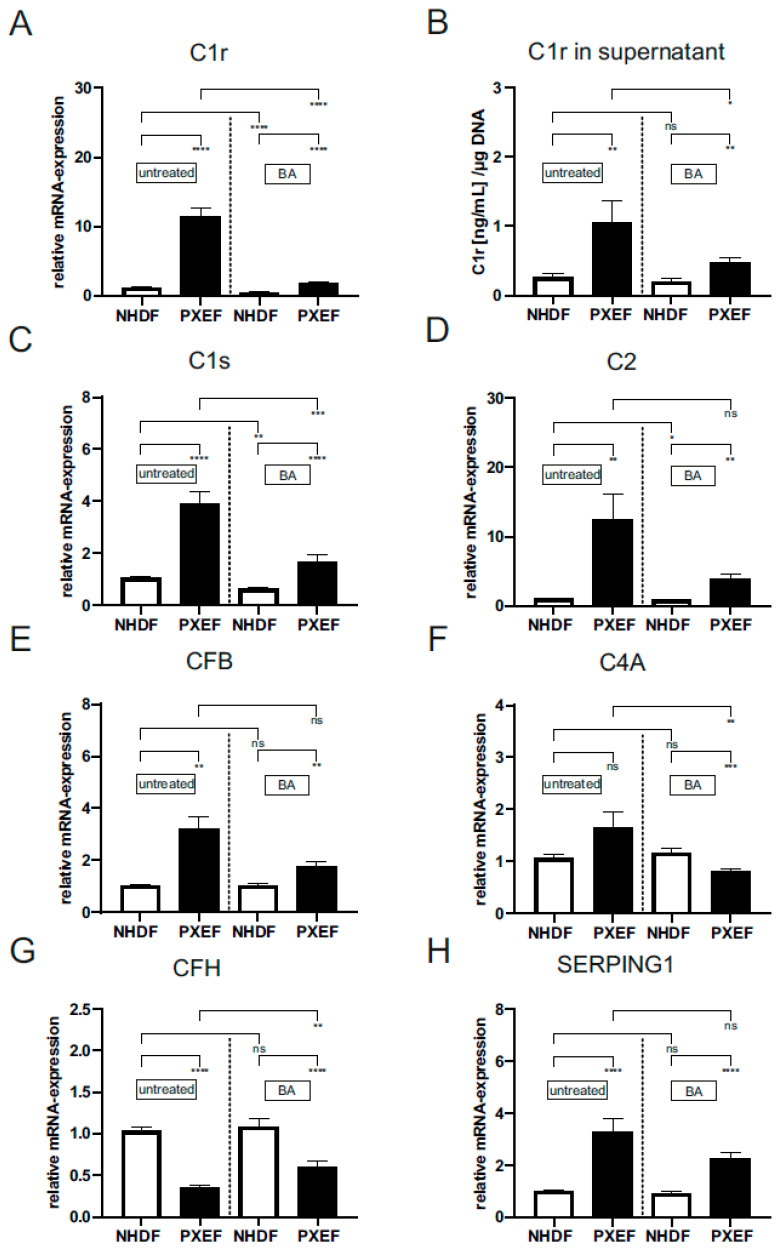
Relative mRNA expression of *C1r*, *C1s*, *C2*, *CFB*, *C4A*, *CFH*, *SERPING1* and the protein level of C1r in supernatants of PXE and control fibroblasts. Fibroblasts from PXE patients (PXEF) and healthy control donors (NHDF) were cultivated for 72 h in 10% LPDS medium with DMSO as vehicle (untreated) or 1 µM BA. (**A**) Gene expression and (**B**) protein level of C1r in PXEF (black) (n = 4) and NHDF (white) (n = 4). mRNA expression of (**C**) *C1s*, (**D**) *C2*, (**E**) *CFB*, (**F**) *C4A*, (**G**) *CFH*, (**H**) and *SERPING1* in PXEF (black) (n = 4) and NHDF (white) (n = 4). Data are shown as mean ± SEM. **** *p* ≤ 0.001, *** *p* ≤ 0.002, ** *p* ≤ 0.01, * *p* ≤ 0.05, ns *p* > 0.05.

**Figure 5 biomedicines-11-02673-f005:**
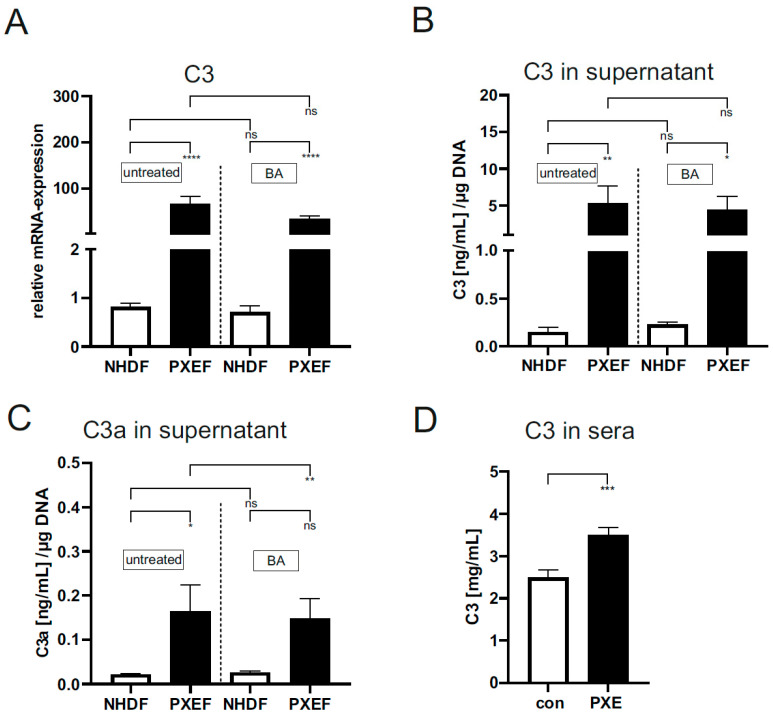
Relative mRNA expression of *C3* and protein concentration of C3 and C3a in supernatants of PXE and control fibroblasts and sera of PXE patients and healthy controls. Fibroblasts from PXE patients (PXEF) and healthy control donors (NHDF) were cultivated for 72 h in 10% LPDS medium with DMSO as vehicle (untreated) or 1 µM BA. The gene expression of (**A**) *C3* and protein level of (**B**) C3 and (**C**) C3a in PXEF (black) and NHDF (white). (**D**) C3 protein concentration in sera from PXE patients (black) (n = 43) and control (white) (n = 41). Data are shown as mean ± SEM. **** *p* ≤ 0.001, *** *p* ≤ 0.002, ** *p* ≤ 0.01, * *p* ≤ 0.05, ns *p* > 0.05.

**Table 1 biomedicines-11-02673-t001:** Characteristics of human dermal fibroblasts from PXE patients and healthy controls.

Sample ID	Gender	Age ^1^	Biopsy Source		*ABCC6* Genotype ^2^	Genotype Status	Phenodex Score ^3^
		**PXE Patients**					
**P3M ^a^**	Male	57	Neck	c.3421C > T (p.Arg1141*)	c.3883_6G > A (SSM)	cht	S3, V2, C0
**P128M ^a^**	Male	51	Neck	c.3769_ 3370insC (p.Leu1259fs*18)	c.3769_3770insC (p.Leu1259fs*18)	hm	S2, E2, G0, C1
**P255F ^a^**	Female	48	Arm	c.3421C > T (p.Arg1141*)	c.2787 + 1C > T (SSM)	cht	S3, E2, G0, C0
**P265F ^a^**	Female	62	Neck	c.1132C > T (p.Gln378*fs)	c.3421C > T (p.Arg1141*)	cht	S3, E3, G0, V1, C1
		**Healthy Controls**					
**M57A ^b^** **(AG13145)**	Male	57	Arm	-	-	wt	None
**M52A ^b^** **(AG11482)**	Male	52	Arm	-	-	wt	None
**F48A ^b^** **(AG14284)**	Female	48	Arm	-	-	wt	None
** *F63A * ^b^ ** ** *(AG12786)* **	Female	63	Arm	-	-	wt	None

cht: compound heterozygous; hm: homozygous; wt: wild type; SSM: splice site mutation. ^a^ Fibroblasts isolated from skin biopsies. ^b^ Fibroblasts purchased from the Coriell Institute for Medical Research (Camden, NJ, USA). ^1^ Age in years. ^2^ Nucleotide numbering refers to the cDNA sequence with the A of the ATG translation initiation start site as nucleotide +1 (GenBank accession number NM_001171.2). ^3^ Adapted from the Phenodex score (an internationally standardized scoring system for the uniform evaluation of PXE cases) according to Legrand et al. [38]. S: skin; E: eye; G: gastrointestinal; V: vascular; C: cardiac.

**Table 2 biomedicines-11-02673-t002:** qPCR protocol for gene expression analysis.

Step	Time [s]	Temperature [°C]	Cycle
Preincubation	300	95	1
Denaturation	10	95	
Annealing	15	T_a_ *	40
Elongation	20	72	
Detection	-	72	
Melting curve	5	95	
	60	65	1
Cooling	60	40	1

* T_a_: primer-specific annealing temperature.

## Data Availability

All data generated or analyzed during this study are available from the corresponding author on request.

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
