# Peer review of "The Activation of JAK/STAT3 Signaling and the Complement System Modulate Inflammation in the Primary Human Dermal Fibroblasts of PXE Patients"

_biomedicines, 2023, doi:10.3390/biomedicines11102673_

Round 1
Reviewer 1 Report
This study investigated activated STAT3 and influence of JAK1/2-inhibitor baricitinib (BA) on inflammatory process and found that a link between JAK/STAT3 signaling and complement activation contributed to the proinflammatory phenotype in PXE fibroblasts. This study is interesting! The experiments were adequately done and interpretation of their findings seemed to be adequate. However, there are several problems to be solved.
(1) In th abstract(line 22-23), “and” should be added before “partly” in “Supplementation with BA reduces JAK/STAT3 activation partly reduces inflammation as well as gene expression”.
(2) In figure 1C, Stat3 protein expression should be examined and shown in each group.
Reviewer 2 Report
General comments
This article studies the involvement of the JAK/STAT3 Signaling pathway and the Complement System in Primary Human Dermal Fibroblasts of PXE Patients in comparison to healthy controls for a better understanding of inflammatory processes associated to this rare disease. Authors hypothesized that JAK/STAT3 signaling is activated and the complement system might play a role in PXE pathogenesis. The introduction is clear and updated. It presents to readers a general landscape of the explored issue. It is a little bit surprising that the protein mutated in this disease, ABCC6 is used for classification at Table 1, but it is not one of the protein/gene followed during most of the the experimental design. The substrate of this ATPase is unknown, but the ABCC6 species in different conditions would be interresing. This absence would be justified.
Anyway, the approach to the problem is plausible. Authors verify the high levels of the JAK/STAT3 signaling pathway in PXE patients and they blocks that pathway by the specific JAK1/2 inhibitor baricitinib (BA) on human dermal fibroblasts. The effect of BA was determined by measuring the mRNA-expression of factors which are involved in the generation of SASP, such as IL-6, MCP-1 and CXCL1, as well as the protein concentration of IL-6 and MCP-1. The section termed “Experimental design” is a brief summary of the approaching techniques and protocols used. This section is not usual in the format of the articles, but I think that it is informative and useful.
Minor points to be addressed.
Line 121: It is established the number of samples/PXE patients (n = 43, 30 female and 13 males, age 44.4 ± 13.5 years and ) and controls (n = 41, 29 female and 12 males, age 44.4 ± 12.9 years) but Table 1 just shows 4 patients and 4 healthy persons. Please, clarify the discrepancies between the number of samples and the partial information provided in Table 1.
Line 154: The abbreviation LPDS should be defined in an abbreviation list or the first time it appears. It is defined at the legend of Figure 1, but this is not convenient for a comprehensive reading. Same applies to baricitinib (BA) defined also at Figure 1, but mentioned as abbreviation before that Figure. I recommend the introduction of an abbreviation list as there are more cases that are used throughout the text and defined elsewhere, but difficult to find (i.e. PXEF, defined as Fibroblasts from PXE patients at the legend of Figure 3).
Line 176: Replace glycerinaldehyde-3-phosphate-dehydrogenase (GAPDH) by the correct name, glyceraldehyde-3-phosphate-dehydrogenase.
Clarify the position and meaning of the brackets located at the middle of Table 2 (around the PCR cycles column)
Line 287: The expression (4-5 ± 1.81) without units is vague. Does it mean 4.5± 1.81?. Units should be provided. For instance, at “pSTAT3 in PXEF by 98% (0.1 ± 0.02)”, the % is ok, but 0.1 should be expressed in the same units, as the 4.5 and others. It seems that they are arbitrary units (intensity/area ratio at the Figures), but even so, this point should be clarified and defined at the beginning. This applies to many other data included in the paragraph (lines 287-301), and subsequent paragraphs. Just as a personal opinion, it seems to me that there is an abuse of the relative expression levels of the different proteins and biochemical markers throughout the manuscript. The % of variations and the bars in the Figures are enough to avoid such a tedious way of expressing relative expression.
Conclusion at lines 588-594 is correct, and it would be under the heading conclusion to clear up the contribution of this study.
